# Tuberous Sclerosis, Type II Diabetes Mellitus and the PI3K/AKT/mTOR Signaling Pathways—Case Report and Literature Review

**DOI:** 10.3390/genes14020433

**Published:** 2023-02-08

**Authors:** Claudia Maria Jurca, Kinga Kozma, Codruta Diana Petchesi, Dana Carmen Zaha, Ioan Magyar, Mihai Munteanu, Lucian Faur, Aurora Jurca, Dan Bembea, Emilia Severin, Alexandru Daniel Jurca

**Affiliations:** 1Department of Preclinical Disciplines, Faculty of Medicine and Pharmacy, University of Oradea, 410081 Oradea, Romania; 2Regional Center of Medical Genetics Bihor, County Emergency Clinical Hospital Oradea (Part of ERN-ITHACA), 410469 Oradea, Romania; 3Department of Medical Disciplines, Faculty of Medicine and Pharmacy, University of Oradea, 410081 Oradea, Romania; 4Faculty of Medicine and Pharmacy, University of Oradea, 410081 Oradea, Romania; 5Faculty of Medicine, University of Medicine and Pharmacy ”Iuliu Hațieganu”, 400012 Cluj Napoca, Romania; 6Department of Genetics, University of Medicine and Pharmacy ”Carol Davila”, 020021 Bucharest, Romania

**Keywords:** tuberous sclerosis, hamartomas, TSC1, TSC2, epilepsy, type 2 diabetes mellitus, Metformin

## Abstract

Tuberous sclerosis complex (TSC) is a rare autosomal dominant neurocutaneous syndrome. It is manifested mainly in cutaneous lesions, epilepsy and the emergence of hamartomas in several tissues and organs. The disease sets in due to mutations in two tumor suppressor genes: *TSC1* and *TSC2*. The authors present the case of a 33-year-old female patient registered with the Bihor County Regional Center of Medical Genetics (RCMG) since 2021 with a TSC diagnosis. She was diagnosed with epilepsy at eight months old. At 18 years old she was diagnosed with tuberous sclerosis and was referred to the neurology department. Since 2013 she has been registered with the department for diabetes and nutritional diseases with a type 2 diabetes mellitus (T2DM) diagnosis. The clinical examination revealed: growth delay, obesity, facial angiofibromas, sebaceous adenomas, depigmented macules, papillomatous tumorlets in the thorax (bilateral) and neck, periungual fibroma in both lower limbs, frequent convulsive seizures; on a biological level, high glycemia and glycated hemoglobin levels. Brain MRI displayed a distinctive TS aspect with five bilateral hamartomatous subependymal nodules associating cortical/subcortical tubers with the frontal, temporal and occipital distribution. Molecular diagnosis showed a pathogenic variant in the TSC1 gene, exon 13, c.1270A>T (p. Arg424*). Current treatment targets diabetes (Metformin, Gliclazide and the GLP-1 analog semaglutide) and epilepsy (Carbamazepine and Clonazepam). This case report presents a rare association between type 2 diabetes mellitus and Tuberous Sclerosis Complex. We suggest that the diabetes medication Metformin may have positive effects on both the progression of the tumor associated with TSC and the seizures specific to TSC and we assume that the association of TSC and T2DM in the presented cases is accidental, as there are no similar cases reported in the literature.

## 1. Introduction

Alongside neurofibromatosis type I and II, von Hippel–Lindau syndrome and Sturge–Weber syndrome, the tuberous sclerosis complex belongs to the larger group of neurocutaneous syndromes. It is a disease that affects multiple organs and systems and is typified by the presence of cutaneous changes, epilepsy and hamartomas in several tissues and organs (brain, myocardium, teguments, kidneys, eyes) [1]. The prevalence of the disease is between 1 in 6000 to 1 in 10,000 live newborns [2,3]. The disease was first described in 1862 by anatomopathologist Friedrich Daniel von Recklinghausen, who identified the presence of cardiac myomas and sclerotic changes in the brain of a newborn who died immediately after birth. Later, in 1880, neurologist Désiré-Magloire Bourneville defined the disease much more clearly, and it has later come to be known by her name: Bourneville tuberous sclerosis [4]. It has no specific symptoms, and its main manifestations are neurological: convulsive seizures that occur early, in the first year of life with most patients, cortical tubers, subependymal hamartomas, and sometimes subependymal astrocytomas. The cutaneous manifestations include depigmented macules and shagreen patch. In adults, renal angiomyolipomas may appear. Behavioral disorders on the ADHD (attention-deficit/hyperactivity disorder) and autism spectrum are also frequently reported [5]. The determinism of the disease involves mutations in two genes, *TSC1* and *TSC2*, both belonging to TSC included in the PI3K/AKT/mTOR signaling pathway, which plays a part in cell growth and proliferation. The tuberous sclerosis complex acts as a brake in the mTOR (mammalian target of rapamycin complex) protein kinase activation cascade; mutations in this complex lead to uncontrolled activation of mTOR, which will in turn determine marked cell growth and proliferation and the development of tumors. Treatment with mTOR inhibitors (Sirolimus, Everolimus) has a positive effect on subependymal astrocytomas, renal angiomyolipomas, and also on facial angiofibromas.

Type 2 diabetes mellitus is a disease with a multifactorial polygenic determinism, as it involves both genetic and environmental factors. It is a chronic disease in which cellular resistance to insulin sets in, followed by a progressive decrease in insulin, causing the incorrect use of glucose, hyperinsulinism and lipid metabolism alterations (dyslipidemia). The genetic component plays an important part in the etiology of T2DM, as proved by studies on twins and families, with over 100 susceptibility loci described to date [6,7,8]. Treatment for T2DM is medical and the progress made in the past 10 years in the development and emergence of antihyperglycemic agents has been remarkable [9,10].

Research on the association between type 2 diabetes and tuberous sclerosis complex is relatively limited. However, several case reports have been published on this topic that suggest that individuals with TSC may have an increased risk of developing type 2 diabetes [11]. This association may be because TSC is caused by mutations in certain genes that play a role in insulin signaling and glucose metabolism. When mutations occur in TSC1 or TSC2 genes, the hamartin-tuberin complex does not function correctly, leading to the activation of mTORC1 and the development of TSC tumors and other symptoms. Regarding glucose metabolism, the mTOR signaling pathway also regulates insulin sensitivity and glucose uptake in cells, and its dysfunction can lead to insulin resistance and type 2 diabetes. Studies have shown that the mTOR pathway is altered in the liver, muscle, and adipose tissue of individuals with TSC, which contributes to the development of type 2 diabetes in this population. Moreover, it has been suggested that the antidiabetic drug Metformin may have beneficial effects in TSC patients, as it has been reported to reduce the size of brain tumors and improve seizure control in some studies. However, more research is needed to fully understand the relationship between TSC and type 2 diabetes, and the potential effects of Metformin in this patient population [12].

This article sets out to present a case of tuberous sclerosis 1 (MIM≠191100) with a pathogenic variant yet not recorded in databases, in a patient associating T2DM, under Metformin treatment and the authors assume that the association of TSC and T2DM in the presented cases is accidental. 

## 2. Materials and Methods

### 2.1. Case Presentation

The case, a 33-year-old woman, was referred to RCMG in 2021 by the neurologist. She is the fourth child in the family (the patient has three healthy siblings). Family history: her mother was diagnosed at age of 30 with type 2 diabetes mellitus, treated initially with oral medication and currently with insulin. Past medical history: the patient has been registered with the pediatric neurology department since she was diagnosed with epilepsy at the age of 8 months. At the age of 18 she was clinically diagnosed with tuberous sclerosis, and at 24 with T2DM.

### 2.2. Laboratory Investigations

The main lab tests assessed glucose metabolism (glycemia, glycated hemoglobin), lipid metabolism (cholesterol, lipids, triglycerides), protein (proteinemia, serum protein electrophoresis), mineral status (calcemia, phosphatemia, magnesemia), hormonal level, etc.

### 2.3. Molecular Investigations

Written informed consent was obtained from the mother prior to participation in the study. DNA was extracted from peripheral blood lymphocytes of the patient using standard extraction procedures. The patient was tested using next-generation sequencing (NGS) with a multi-gene panel of two genes, *TSC1* and *TSC2*. The targeted regions were sequenced with ≥50× depth or were supplemented with additional analysis. After high-throughput sequencing using Illumina technology, the output reads were aligned to a reference sequence (genome build GRCh37; custom derivative of the RefSeq transcriptome) to identify the locations of exon junctions through the detection of split reads. The relative usage of exon junctions in a test specimen was assessed quantitatively and compared to the usage seen in control specimens. Abnormal exon junction usage was evaluated as evidence in the Sherloc variant interpretation framework. If an abnormal splicing pattern was predicted based on a DNA variant outside the typical reportable range, the presence of the variant was confirmed by targeted DNA sequencing. Enrichment and analysis focus on the coding sequence of the indicated transcripts, 20 bp of flanking intronic sequence and other specific genomic regions demonstrated to be causative of disease at the time of assay design. Promoters, untranslated regions, and other non-coding regions were not otherwise interrogated. For some genes only targeted loci were analyzed. Exonic deletions and duplications were called using an in-house algorithm that determines copy number at each target by comparing the read depth for each target in the proband sequence with both mean read-depth and read-depth distribution obtained from a set of clinical samples.

## 3. Results

### 3.1. Clinical Evaluation of the Patient

The patient presents facial dysmorphism: round facies, double chin, facial cutaneous angiomas and more extended sebaceous adenomas in the malar area, the cheekbones, nasolabial folds, and nasal pyramid; high forehead, deep set eyes, thin lips; excessive subcutaneous tissue in thorax, abdomen and limbs, with an abdominal skinfold of 5–6 cm; weight: 90 kg (+2 SD), height: 162 cm (percentile 50–25) (Figure 1A,B).

Multiple depigmented macules in the neck and interscapular area; papillomatous tumorlets in the thorax (bilateral) and neck, (Figure 1C), dental caries, sublingual fibroma, oligodontia and tooth enamel dystrophy (Figure 2A,B).

On the lower limbs, large ungual fibromas that were surgically removed (Figure 3A,B). Cardiology examination shows a low tolerance for effort, sinus tachycardia, I/II-degree tricuspid insufficiency and normal electrocardiogram (ECG).

The patient displays intellectual disability: delayed standing and walking, profound language learning difficulties, nervousness, irritability and convulsive seizures, and she has been registered with the neurology department since infancy.

### 3.2. Laboratory Investigations

Table 1 shows the laboratory investigations performed on the patient in the past 18 months. In this table we can also note the descending trend in glycemia and glycated hemoglobin values; triglyceride values have remained consistently high.

### 3.3. Radiology Investigations

The abdominal ultrasound showed multiple small vesical calculi and hepatic steatosis. The last brain MRI (2022) displayed an STC-suggestive aspect, with four bilateral hamartomatous subependymal nodules—the largest 3 mm in diameter, partially calcified—associating cortical/subcortical tubers with frontal, temporal and occipital distribution on the right and frontal and parieto-occipital distribution on the left. There were no pathological and evolutionary changes when compared to previous MRIs (in the past 6 years) (Figure 4).

### 3.4. Molecular Investigations

The sequence analysis and deletion/duplication testing of the two genes *TSC1* and *TSC2* show a pathogenic variant in the *TSC1* gene, exon 13, c.1270A>T (p. Arg424*), heterozygous. This sequence change creates a premature translational stop signal (p. Arg424*) in the *TSC1* gene.

## 4. Discussion

### 4.1. PI3K/AKT/ TSC1/TSC2/mTOR Pathway

This is an intracellular signaling pathway that involves numerous biological processes, the most important of which are cellular proliferation, apoptosis and angiogenesis, but it also plays a part in regulating glucose metabolism. The cascade is initiated by the binding of extracellular growth factors (epidermal growth factor—EGF, insulin-like growth factor IGF, fibroblast growth factor—FGF, etc.) to the transmembrane receptor tyrosine kinases (RTKs). This leads to the activation of phosphatidylinositol 3-kinase (PI3K), which, in turn, activates the serine/threonine protein kinase B, AKT (also called PKB), that will phosphorylate downstream other molecules, such as the TSC complex, which, in turn, will inhibit downstream the mTOR complex (mechanistic target of rapamycin). The mTOR is, in fact, a serine/threonine kinase belonging to the PI3K-related kinase family, with an important role in cellular growth and proliferation [13].

#### 4.1.1. PI3K

According to structure and specific substrate, PI3K is divided into three classes: I, II and III. Of these, class I is the most studied and it is subdivided into subclasses IA and IB.

Class IA is activated by RTKs and G protein-coupled receptors (GPCR). Class IB is activated only by GPCR [14]. Class IA consists of the regulating element, subunit p85, and the catalytic element, subunit p110. By binding p85 to RTKs on the surface of the cellular membrane, P13K will be activated; then the catalytic subunit p110 will become activated and will catalyze the formation of the cellular second messenger PIP3. In its turn, PIP3 will recruit PDK1 and AKT kinase which will activate other molecules downstream [15,16,17].

Class IB contains the regulating subunit p101 and the catalytic subunit p110γ and the activation of this pathway is realized by the direct binding of p110γ to GPCR.

Although the role that PI3P plays in cellular growth, differentiation and apoptosis is well known, the precise function of its subunits in glucose metabolism and transport is not fully known at the moment [18,19].

#### 4.1.2. AKT

AKT is the main mediator in the PI3K signaling pathway, situated downstream from it, and it is a serine/threonine kinase that is subdivided into 3 types: AKT1, AKT2 and AKT3, [20,21]. Of these subunits, AKT1 is expressed in various tissues, AKT2 is only expressed in insulin-dependent tissues, and AKT3 is only expressed in cerebral and testicular tissues; through this wide expression in a multitude of tissues, AKT contributes to maintaining the tissues’ normal physiological function. Once activated, AKT will phosphorylate other molecules downstream, and more precisely it will activate the TSC complex [22,23].

#### 4.1.3. The TSC Complex

The TSC complex consists of two genes: *TSC1* and *TSC2*, alongside TBC1D7 (Tre2-Bub2-Cdc16-1 domain family member 7). The complex plays an important part in inhibiting the signals in the mTOR/S6K/4E-BP pathway—more precisely, it inhibits the activity of Ras homolog enriched in the brain (Rheb), which will no longer be able to activate mTOR downstream [24,25]. If the TSC1-TSC2 complex loses its function, the mTOR complex is activated, and control over cellular growth mechanisms is lost [26,27].

#### 4.1.4. mTOR

Activated AKT phosphorylates the TSC1-TSC2 complex, which will inhibit the mTOR complex downstream through Rheb [28]. In its turn, mTOR further phosphorylates other molecules, 4EBP1 and S6Ks, that are part of the complex initiating protein translation, YY1 respectively, involved in mitochondrial respiration [29,30]. The mTOR-S6K complex will negatively regulate upstream the insulin receptor substrate pathways (IRS) by direct phosphorylation of serine/threonine residues [31,32,33]. This hyperphosphorylation of residues leads to insulin resistance associated with overstimulation of mTOR. mTOR consists of the mTORC1 complex and the mTORC2 complex, each with its distinct components [34,35].

If the mechanisms activating mTORC1 are well-known, little is known about the activation of mTORC2. It responds to growth factors by directly associating with the ribosomes, independently of the PI3K pathway. After activation, mTORC2 will directly activate AKT [36]. mTORC1 has multiple roles—it is involved in energy metabolism through the production of ATP, and it stimulates protein synthesis and lipid synthesis, but it also plays a role in gluconeogenesis when activated, as it is responsive to rapamycin, while mTORC2 is not responsive to rapamycin and it is involved in the cytoskeletal dynamic. It is a well-known fact that mTORC1 feels and integrates both intra- and extra-cellular signals, while only growth factors stimulate the activity of mTORC2 [37,38]. The main components of the PI3K/AKT/mTOR signaling pathway are represented in Figure 5.

The mTOR pathway can have anti- and pro-diabetic effects depending on the specific cell type and physiological context. In TSC, the dysfunction of the TSC1 and TSC2 genes leads to the activation of mTORC1 and causes cellular growth and proliferation. This can lead to the development of TSC-associated tumors and other symptoms, but also metabolic dysregulation. In addition, it’s also been proposed that mTORC1 hyperactivity may modulate the inflammatory response, which also plays a role in T2DM. However, further research is needed to fully understand the underlying mechanisms and how mTOR activity affects diabetes in TSC [39]. The association of the mTOR pathway with type 2 diabetes mellitus (T2DM) is a complex and active area of research. Studies have shown that the mTOR pathway is activated in the liver, muscle and adipose tissue of individuals with T2DM, which can lead to insulin resistance and decreased glucose uptake in cells [40,41]. Postprandial increase in blood sugar levels will activate the mTOR complex and, consequently, also the AKT protein (directly by mTORC2); the activation of AKT will stimulate the glucose depositing in adipocytes and will increase glycogen activity and its buildup in the liver and muscles [40,41].mTORC1 responds to insulin and regulates glucose metabolism in the liver by controlling the activity of the transcription factor FOXO1, which regulates the expression of genes involved in glucose metabolism. The activation of mTORC1 in the liver leads to the inhibition of FOXO1 and the decreased expression of genes involved in glucose metabolism, leading to hyperglycemia. In muscle, mTORC1 activation also leads to insulin resistance by decreasing the expression of the insulin receptor substrate-1 (IRS-1) and GLUT4, which are required for insulin-stimulated glucose uptake. mTORC1 can also affect glucose metabolism by modulating autophagy, a process that helps cells recycle old and damaged proteins. In adipose tissue, mTORC1 activation leads to the inhibition of the transcription factor PPARγ, which regulates the expression of genes involved in lipid metabolism, leading to the accumulation of lipids in the adipose tissue and the development of insulin resistance [42,43]. Selman C et al. have shown that S6K1 depletion improves insulin sensitivity in mice, while the activation of the mTOR complex produces dysfunctions in the insulin signaling pathway and insulin resistance sets in [44,45]. In summary, the mTOR pathway plays a critical role in the regulation of glucose metabolism in various tissues and its activation in T2DM can lead to insulin resistance, decreased glucose uptake, and hyperglycemia. More research is needed to fully understand the underlying mechanisms of mTOR association with T2DM and how these mechanisms differ from those in TSC.

### 4.2. Clinical Aspects

According to the latest review of the Sclerosis Complex Consensus in 2021, the criteria for diagnosing TSC are divided into major and minor and are illustrated in Table 2. For a positive diagnosis, it is necessary to have two major criteria and one or two minor criteria present or to have the mutation identified in one of the two genes [46].

The case presented meets the clinical criteria for classification as TSC: depigmented macules, facial angiomas, sublingual and periungual fibromas, changes in tooth enamel, cortical tubers and subependymal hamartomatous nodules. The patient also manifests convulsive seizures that started in infancy (8 months old) as infantile spasms, in congruence with the data in the literature regarding the start of convulsions in early childhood [47,48]. Starting at age 10, there was a marked and accelerated weight gain, with the patient reaching a weight of 120 kg in 2021, at age 32.

TSC patients regularly have behavioral and psychiatric problems, and a high percentage shows signs of ADHD or autism, depression, and anxiety. The association between the intensification of behavioral disorders and the early debut of epilepsy has been intensely studied in the literature, as there is a direct connection between the two criteria [49,50]. The early debut and the severity of the convulsions affected the patient’s neurobehavioral development; she said her first words after 24 months and completed the first eight grades in a special needs school. Her behavior is infantile, and she has nervousness episodes. She doesn’t like stepping out of her daily routine, she dislikes change. 

### 4.3. Genetics

The tuberous sclerosis complex appears as a consequence of mutations in the two genes, *TSC1* and *TSC2*, identified in 1990 on Drosophila [51].

The *TSC1* gene is located on the long arm of chromosome 9 (9q34), it contains 23 exons and encodes a protein named hamartin, with a molecular weight of 130kd, made up of 1164 amino acids. 

The *TSC2* gene is located on the short arm of chromosome 16 (16p13.6), it contains 41 exons and encodes a protein called tuberin, located in the Golgi apparatus, with a role in vesicular transport [52]. The locus of the TSC2 gene is tightly connected to the locus of the PKD gene, the mutations of which cause autosomal dominant polycystic kidney disease. If the two genes are affected, a contiguous gene syndrome sets in—infantile severe polycystic kidney disease, with tuberous sclerosis PKD-TSC (MIM #600273).

Identification of the mutation in the two genes is sufficient to confirm diagnosis and 75–90% of the tested patients have a positive molecular result for one of the two genes. A negative molecular result does not exclude a TSC diagnosis, as there can be somatic mosaicism, or mutations present in the promoter region or the introns, an aspect present in 10–15% of TSC patients [53]. The mutations in the two genes may be de novo—in two-thirds of the cases, or inherited—in one-third of the cases, in an autosomal dominant pattern of inheritance [54,55].

Tuberous sclerosis is a disease with 100% penetrance but with variable expressivity and a large range of mutations described in the literature. The mutations occurring in *TSC1* are usually nonsense or frameshift mutations, leading to the formation of a truncated protein or the absence of protein, while mutations in gene *TSC2* are often missense, with large deletions and rearrangements [56,57]. The large majority of people with TS exhibit mutations in *TSC2* (70–90%), but only 10–20% of those affected have mutations in *TSC1*, most often in exons 15–17. Numerous pathogenic variants are described in the databases, both in *TSC1* and in *TSC2*, as more and more pathogenic variants are being identified due to the increased availability and affordability of genetic testing.

In the case of our patient, the mutation is in gene *TSC1*, exon 13 being a pathogenic variant, c.1270A>T (p. Arg424*). This variant is not present in population databases and has not been reported in the literature in individuals affected in *TSC1*.

### 4.4. Treatment

Tuberous sclerosis does not have a disease-specific treatment—patients receive symptomatic treatment only. As it is a multisystem disease, with no disease-specific symptoms, the management thereof calls for the presence of a multidisciplinary team involving genetics, neurology, dermatology, internal medicine, surgery, ophthalmology, pneumology, cardiology, psychology and dentistry [58,59]. Thus, intensely vascularized facial angiomas are treated with intense pulsed light (IPL), ungular fibromas are surgically removed, and convulsions are stopped with anti-epileptic medication.

#### 4.4.1. Treatment with mTOR Inhibitors

Using mTOR inhibitors has become the basic treatment of tumors in TS. The best-known inhibitor of the mTOR pathway is Rapamycin (Sirolimus) and its derivative, Everolimus, but Metformin, an oral antidiabetic, also inhibits mTOR [60].

##### Treatment with Rapamycin

In TSC, as the mTOR pathway is disrupted, an uncontrolled activation thereof occurs, resulting in an intense activity of cellular division and proliferation. As a result of this proliferation, the angiogenesis process is intensified, which leads to the development of cutaneous and renal angiofibromas throughout the patient’s life, but also of renal angiomyolipomas, whose pathophysiology includes an accumulation of fat. Fat accumulations are also found in the myocardium of TS patients [61].

The fact that *TSC1* and *TSC2* play an important part in regulating the mTOR signaling pathway, embedding growth messages and coordinating them downstream to mTOR, has led to the use of mTOR inhibitors (Rapamycin) in the treatment of TSC. Rapamycin is a macrolide compound that was first identified in Chile, on the island of Rapa Nui (Easter Island) in 1965, in Streptomyces hygroscopicus. By binding mTOR, it inhibits its activity, stopping uncontrolled cellular growth [62]. Initially, it was used as an antifungal. Later on, starting in 2006, Rapamycin (Sirolimus) and its derivative, Everolimus, became subjects of intense research that clearly proved their immunosuppressant and anticancer role [63,64]. Although Sirolimus and Everolimus have the same molecular mechanism, they have different clinical profiles.

Sirolimus was approved by the European Medicines Agency (EMA) and by the U.S. Food and Drug Administration (FDA) for the treatment of lymphangioleiomyomatosis (LAM) in TSC patients and for preventing organ rejection, while Everolimus was approved for the treatment of subependymal astrocytomas with surgical contraindication and of renal angiolipomas [65,66].

In Brazil, Agência Nacional de Vigilância Sanitária (ANVISA) approved the use of Everolimus in cases of renal tumors in TSC patients, and of Sirolimus in the treatment of facial angiofibromas [67]. It was noted that the response to Sirolimus was more prompt in the case of facial cutaneous angiofibromas than in ungular fibromas and subependymal astrocytomas, which did not respond as favorably; this seems to be due to the anti-angioproliferative effect of Sirolimus (it is very efficient in intensely vascularized facial tumors). There was no record of tumor rebound after stopping treatment, although most studies demonstrated that discontinuing the treatment will lead to tumoral relapse [66,68,69].

Treatment with mTOR inhibitors is associated with various adverse reactions, most frequently with the disruption of glucose metabolism [70], but also with stomatitis and infections. Different clinical trials demonstrated the presence of hyperglycemia in 13–50% of oncological patients who had been administered this treatment with mTOR inhibitors [71]. Chakrabarti et al. demonstrated that treatment with Rapamycin in transplant patients has as a side effect dyslipidemia [72].

##### Treatment with Metformin

The biguanide known as Metformin is an oral antidiabetic that also plays a key role in the mTOR pathway, through the AMPK serine-threonine kinase. In low energy conditions, Metformin inhibits the mitochondrial complex I and activates the AMPK, which is able to detect the energy level inside the cell, stopping cellular growth when it is low [73,74]. In order to balance the energy level, AMPK phosphorylates several molecules located downstream, stimulating catabolism and inhibiting anabolic ATP-consuming pathways [75,76].

The molecular bases of its therapeutic role have not yet been completely elucidated. As for its involvement and role in tuberous sclerosis, recent studies have shown that there are several ways in which Metformin inhibits the mTORC1 complex: (a) (through AMPK) it determines the activation of *TSC1* and *TSC2* genes; (b) AMPK phosphorylates direct raptor (positive regulator of mTORC1) needed to inhibit mTORC1; (c) by blocking IGF1 and insulin, TSC complex inhibits mTORC1, (both insulin and IGF1 have the role of inhibiting the two genes in order to determine cellular growth); (d) can induce p53, which inhibits mTORC1; (e) amplifies the expression of the *DICER1* gene (encoded enzyme DICER), breaking the RNA and the pre-micro RNA into small bits of micro RNA and RNAi (small interfering RNA), as the mutations in this gene lead to tumoral syndrome. This mechanism highlights the role of Metformin as an antineoplastic agent; (f) it inhibits HIF-1α (hypoxia-inducible factor) which mediates the response to low tissular oxygen levels through AMPK and mTORC; (g) inhibits fatty acids synthase, which catalyzes the hepatic fatty acids synthesis; and (h) inactivates direct regulators, thus inhibiting mTORC1 by unbinding the latter from its Rheb activator [77,78,79]. In addition, as an antineoplastic agent, Metformin inhibits c-MYC, which is a proto-oncogene over-expressed in certain tumor forms [80,81]. The main effects of Metformin are synthetically represented in Figure 6.

Some studies have found that Metformin does not reduce tumor volume or seizure frequency when compared to a placebo, while other studies have found that it does. In a multicenter randomized double-blind trial, Amin et al. administered Metformin and placebo to 51 patients with tuberous sclerosis for 1 year. The study followed the evolution of angiomyolipomas, subependymal astrocytomas and epileptic seizures in patients who were treated with Metformin as compared to those treated with placebo medication. The study concluded that the volume of both (renal angiomyolipomas and subependymal astrocytomas) decreased significantly and the epileptic seizures were much less frequent in patients treated with Metformin; there were only three major adverse reactions reported in the group treated with this medicine. The authors specified that Metformin is a safe and well-tolerated drug (both in children and adults with TS) [82,83].

Our patient did not receive treatment with mTOR inhibitors like Everolimus or Sirolimus, as she did not have the medical recommendations for it. Her treatment aimed at controlling diabetes and epilepsy, as well as symptomatically when was necessary.

The authors can explain the inconsistent results by acknowledging that the relationship between TSC and T2DM is complex and that the effects of Metformin on both conditions are not fully understood. It is important to note that the conflicting results may be due to differences in study design, patient population, or dosage of Metformin used. We recommend that healthcare providers exercise caution when prescribing Metformin to patients with TSC and diabetes.

#### 4.4.2. Treatment of Diabetes Mellitus and Obesity

The T2DM diagnosis was established at the age of 23 based on the positive family history, the clinical picture (obesity, polyuria, polydipsia, polyphagia), hyperglycemia and glycosylated hemoglobin. No antibody assays were performed for specific type 1 diabetes mellitus (T1DM) antibodies, anti-GAD, anti-insulin, or islet cell autoantibodies. The absence of diabetic ketoacidosis episodes and the relatively good response to oral treatment excluded T1DM. The initial treatment consisted of recommendations regarding the optimization of the patient’s lifestyle and a combination of Metformin hydrochloride and Glibenclamide 2.5/400 mg, 3 tablets/day. During the January 2021 re-evaluation, the glycated hemoglobin value was 8.4%, which is why it was decided to change her treatment to Metformin 2000 mg/day and Gliclazide extended-release tablets (60 mg), 1 tablet/day in the morning. The patient did not follow the recommendations regarding her diet, as she has a compulsive tendency to eat high-calorie foods, such as sweets. In February 2022 an analogous treatment with GLP-1 semaglutide injectable 0.5 mg/week was associated, aiming to reduce appetite and assist in weight loss, thus reducing insulin resistance and improving glycemic control parameters.

The clinical and metabolic evolution was slow but favorable; the patient lost 25 kg in the following 9 months (weight to date—90 kg), glycemia at the time of the latest assessment was at 147 mg/dL and glycated hemoglobin values dropped. The Metformin treatment also seems to have had a favorable effect on the tumoral evolution of TSC, the patient developed no further renal angioleiomyomas or subependymal astrocytomas, and the existing ones were stationary [84].

#### 4.4.3. Treatment for Epilepsy

There is a wide range of anticonvulsants that can be used in TSC. In the case of medication-refractory seizures, there are therapeutic alternatives such as surgical procedures and the ketogenic diet. Despite complex treatment measures, seizures persist in over 60% of patients [85]. In the case presented by the authors, anticonvulsant treatment was initiated during infancy—initially Carbamazepine monotherapy, the current dosage being 1200 mg/day. Clinical evolution was favorable, the seizures becoming less frequent (1–2 every 10 days). In December 2021, affirmatively after a second dose of the anti-COVID-19 vaccine, the patient presented frequent epileptic seizures, especially during the night, which is why treatment with Clonazepam 0.5 mg/day was associated.

### 4.5. Evolution and Monitoring

For full monitoring of the disease, the International Tuberous Sclerosis Complex Consensus group recommends the following annual checkups: dermatology, neuroimaging (MRI for the assessment of tubers and subependymal nodules), EEG (in case seizures occur), dental and ophthalmological (for possible retinal hamartomas). Another factor to be taken into consideration is extended family history—at least over three generations. Currently, our patient is permanently monitored by a complex team of specialists from the following departments: neurology, diabetes and nutritional diseases, genetics, psychology, dentistry and cardiology.

### 4.6. Genetic Counseling

Tuberous sclerosis complex is an autosomal dominant monogenic disease. The recurrence risk in the case of one affected parent is 50% in each descendant. If the parents of the affected person are healthy, we can take into account de novo mutations or germinal mosaicism. Extended family history and genetic pedigree (family tree) are essential for accurate genetic counseling.

## 5. Conclusions

TSC is a disease with a wide range of clinical manifestations. Due to the complexity of the clinical manifestations, proper monitoring requires the involvement of a multidisciplinary team. The case report highlights the association of TSC with T2DM and the potential benefits of Metformin on both the volume of TSC-associated tumors and the frequency of seizures. These findings suggest that individuals with TSC may be at an increased risk of developing T2DM and that Metformin may be a useful treatment option for both T2DM and TSC. However, further research is needed to confirm these findings and to fully understand the underlying mechanisms and the interactions between TSC, diabetes, and Metformin.

## Figures and Tables

**Figure 1 genes-14-00433-f001:**
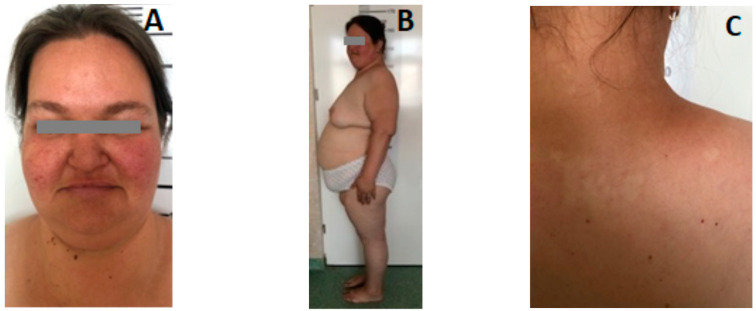
(**A**): Round faces; purplish red facial skin angiomas on the nose, malar and cheekbones; (**B**): lateral view; excess subcutaneous tissue; (**C**) Multiple depigmented macules and papillomatous tumorlets in the neck and thorax.

**Figure 2 genes-14-00433-f002:**
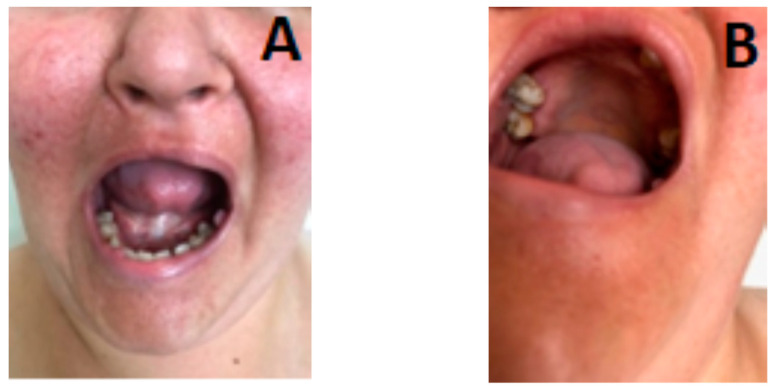
Oro-dental features: (**A**) Dental caries, sublingual fibroma; (**B**) Oligodontia, tooth enamel alteration.

**Figure 3 genes-14-00433-f003:**
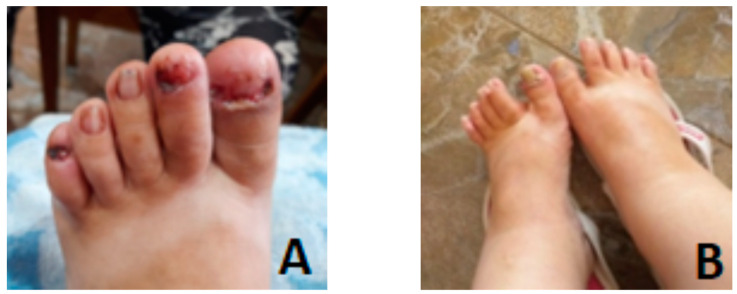
Ungual fibromas: (**A**) after surgery; (**B**) before surgery.

**Figure 4 genes-14-00433-f004:**
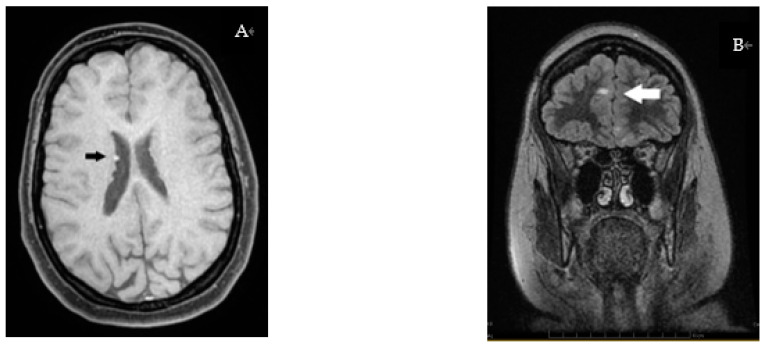
Brain MRI: (**A**) Subependymal nodules; (**B**) Frontal cortico-subcortical tubers.

**Figure 5 genes-14-00433-f005:**
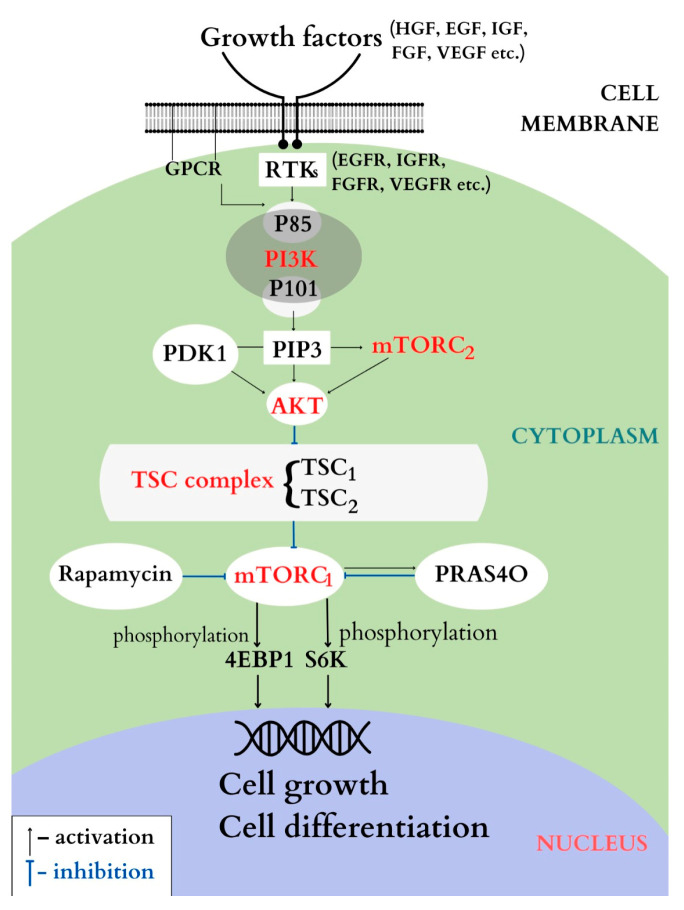
PI3K/AKT/mTOR pathway.

**Figure 6 genes-14-00433-f006:**
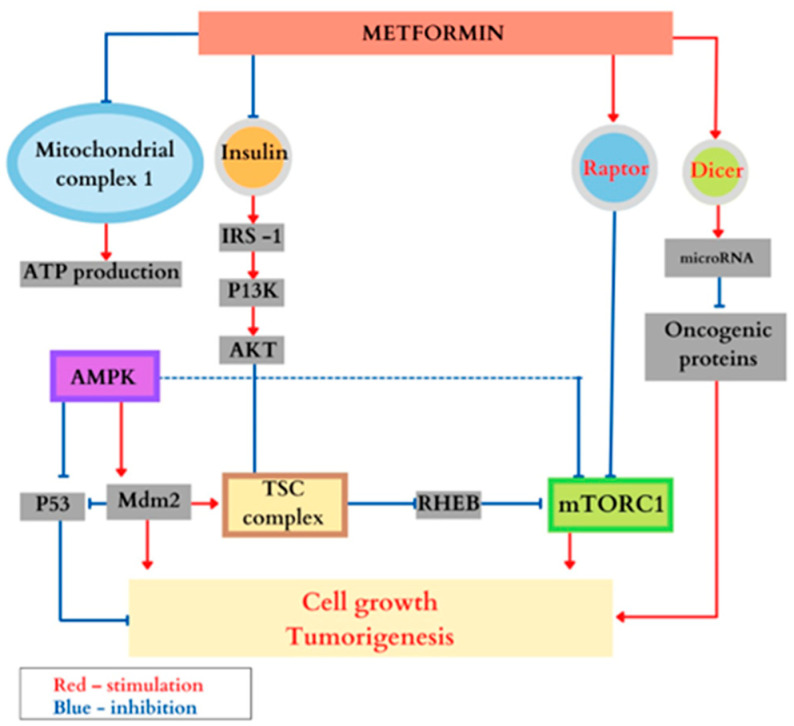
The most important effects of Metformin.

**Table 1 genes-14-00433-t001:** Laboratory Investigations.

Investigations	Jan 2021	July 2021	March 2022	June 2022	August 2022
Glycemia (RV 74–106 mg/dL)	264.7	160	190	178	141
Glycated hemoglobin (RV 4–6%)	8.7	7.6	8.1	7.4	6.5
Cholesterol (RV sub < 200 mg/dL)	212.2	187.6	193.1	178	190
HDL cholesterol (RV 29.8–84.75 mg/dL)	36.45	57.25	65.7	54.6	50
LDL cholesterol (RV 74–106 mg/dL)	142	100	112	99.8	102
Triglycerides (RV < 150 mg/dL)	168	152	158	161	213
Urea (RV 16.8–43.2 mg/dL)	48.5	17.4	22.1	24.5	32.48
Creatinine (RV 0.6–1.2 mg/dL)	1.14	0.67	0.78	0.83	0.58
Glomerular filtration rate * RV > 90; between 60–89 moderate decrease	64.20	117.13	131.62	124.86	122.01

* RV-reference value.

**Table 2 genes-14-00433-t002:** Criteria for a positive tuberous sclerosis diagnosis.

Major Criteria (11)	Observation
Depigmented macules with a 3–5 mm diameter Facial angiofibromas (over 3) Ungual fibromas (over 2) Shagreen patch Retinal hamartomas Cortical dysplasia Astrocytomas and subependymal nodules Cardiac rhabdomyomas Lymphangioleiomyomatosis renal angiomyolipomas	Genetic diagnosis: for a positive diagnosis it is sufficient to identify a pathogenic variant in one of the two genes
**Minor criteria (7)**	
“Confetti” skin lesions Intraoral fibromas Changes in tooth enamel White patches on the retina Renal cysts Other hamartomas Sclerotic bone changes	

## Data Availability

Not applicable.

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
