# Peer review of "Tuberous Sclerosis, Type II Diabetes Mellitus and the PI3K/AKT/mTOR Signaling Pathways—Case Report and Literature Review"

_genes, 2023, doi:10.3390/genes14020433_

Round 1
Reviewer 1 Report
This article attempts to report on the association of the tuberous sclerosis complex with type 2 diabetes and the potential therapeutic role of metformin, which is indeed relatively novel. There are still some important issues that need to be revised.
Abstract
The author's explanation of the conclusion may be too affirmed. The current information is not enough to draw such a conclusion, and it is recommended to modify the expression.
Introduction
It is suggested that the authors should focus more on the research and case reports of type 2 diabetes in the tuberous sclerosis complex.
Materials and methods
Studies have shown that: metformin did not resl volume. Metformin did reda volume and seizure filed with placebo. The author should explain in the text.
Discussions
The authors over-review the pathogenesis of the tuberous sclerosis complex, which is not relevant to the main theme of this paper. Some reductions should be made to focus more on possible mechanisms of diabetes and possible therapeutic mechanisms of metformin. The author's conclusions are not very clear, and the extrapolation of the conclusions seems to be too hasty and not yet well grounded in theory.
Author Response
Please see the attachement.

Reviewer 2 Report
Review for TSC
The authors describe one clinical case of tuberous sclerosis 1 (MIM≠191100) with a pathogenic variant yet not recorded in the TSC database. This pathogenic variant appears to be associated (among other manifestations of TSC) with T2DM. This is of interest for the scientific community as this is a novel pathogenic variant in humans. The finding that there seems to be a link to T2DM is documented by the effects of the anti-glycemic drug metformin in this patient.
Minor points:
The authors should specify as to whether the patient was given sirolimus/everolimus (the standard treatment of TSC) before as sirolimus appears to be contra-indicated in T2DM.
The authors should give a mechanistic explanation (or at least a discussion about) on how increased mTOR activity (in TSC) can lead to T2DM. mTOR is known to function on both pancreatic islet β cells and immune cells and it seems that mTOR has both anti- and prodiabetic effects.
Also shortening the MoA of the PAM (PI3K/AKT/mTOR) (just give refs) in favor of the mTOR association with T2DM would be highly recommended.
Round 2
Reviewer 1 Report
The modified papers can be accepted.